# Diagnostic Accuracy of Physical Examination and Pulse Oximetry for Critical Congenital Cardiac Disease Screening in Newborns

**DOI:** 10.3390/children11010047

**Published:** 2023-12-29

**Authors:** Jari T. van Vliet, Naizihijwa G. Majani, Pilly Chillo, Martijn G. Slieker

**Affiliations:** 1Department of Pediatric Cardiology, Wilhelmina Childrens Hospital, University Medical Center Utrecht, 3584 CX Utrecht, The Netherlands; j.t.vanvliet-7@umcutrecht.nl (J.T.v.V.); n.g.majani@umcutrecht.nl (N.G.M.); 2Department of Pediatric Cardiology, The Jakaya Kikwete Cardiac Institute, Dar es Salaam 65141, Tanzania; 3Department of Internal Medicine, School of Medicine, Faculty of Adult Cardiology, Muhimbili Campus, Muhimbili University of Health and Allied Sciences, Dar es Salaam 65001, Tanzania; pchillo@hsph.harvard.edu

**Keywords:** newborn screening, critical congenital heart disease, pulse oximetry

## Abstract

Background: Newborns with a critical congenital heart disease left undiagnosed and untreated have a substantial risk for serious complications and subsequent failure to thrive. Prenatal ultrasound screening is not widely available, nor is postnatal echocardiography. Physical examination is the standard for postnatal screening. Pulse oximetry has been proposed in numerous studies as an alternative screening method. This systematic review and meta-analysis aims to determine the diagnostic accuracies of both screening methods separately and combined. Methods: A systematic literature search of the Embase, PubMed, and Global Health databases up to 30 November 2023 was conducted with the following keywords: critical congenital heart disease, physical examination, clinical scores, pulse oximetry, and echocardiography. The search included all studies conducted in the newborn period using both physical examination and pulse oximetry as screening methods and excluded newborns admitted to the intensive care unit. All studies were assessed for risk of bias and applicability concerns using the QUADAS-2 score. The review adhered to the PRISMA 2020 statement guideline. Results: Out of 2711 articles, 20 articles were selected as eligible for meta-analysis. Cumulatively, the sample included 872,549 screened newborns. The pooled sensitivity of the physical examination screening method was found to be 0.69 (0.66–0.73 (95% CI)) and specificity was found to be 0.98 (0.98–0.98). For the pulse oximetry screening method, the pooled sensitivity and specificity yielded 0.78 (0.75–0.82) and 0.99 (0.99–0.99), respectively. The combined method of screening yielded improved diagnostic characteristics at a sensitivity and specificity of 0.93 (0.91–0.95) and 0.98 (0.98–0.98, respectively. Conclusions: The evidence indicates that combining both physical examination and pulse oximetry to screen for critical congenital heart disease exceeds the accuracy of either separate method. The main limitation is that solely newborns with suspected critical congenital heart disease were subjected to the reference standard. We recommend adapting both methods to screen for critical congenital heart diseases, especially in settings lacking standard fetal ultrasound screening. To increase the sensitivity further, we recommend increasing the screening time window and employing the peripheral perfusion index.

## 1. Introduction

Congenital Heart Diseases (CHDs) have an incidence of around 9 in every 1000 births. One in four newborns will have a CHD labelled as critical [1,2,3,4,5]. Critical CHDs (CCHDs) are defined as lesions requiring surgery or catheter-based intervention in the first year of life to provide better chances of long-term survival [1]. These newborns may appear healthy after birth but will develop symptoms such as tachypnoea, tachycardia, cyanosis, and hypotension [6]. 

A CCHD is defined as one the following pathologies: Coarctation of the aorta, double-outlet right ventricle, D-transposition of the great arteries, Ebstein’s anomaly, hypoplastic left heart syndrome, interrupted aortic arch, pulmonary atresia, single ventricle, total anomalous pulmonary venous return, tetralogy of Fallot, tricuspid atresia, and persistent truncus arteriosus [1].

Early detection of CCHD is essential for reducing infant morbidity, mortality, and disability [7,8]. In High-Income Countries (HICs), fetal ultrasound is the standard screening method. On average, more than 50% of all CCHD cases are prenatally detected with fetal ultrasound [9]. Standard newborn Physical Examination (PE), such as cardiac auscultation and palpation of femoral pulsations, and Pulse Oximetry (PO) could suggest a remaining undetected CCHD case after birth. In the case of any subsequent suspicion of heart disease, newborn echocardiography is indicated and provides a reference standard diagnosis. Both PE and PO are already in use for hospital births in parts of Europe and the United States [1]. 

However, in Low- or Middle-Income Countries (LMICs), fetal ultrasound screening is not the standard of care due to its unavailability. This means that fetuses with CCHDs in LMICs are often delivered undiagnosed in a homebirth setting [10]. Currently, newborn PE is the sole screening method [11]. To reduce child mortality further, an additional sensitive postnatal screening method is desired. This screening method is required to provide a sufficiently high specificity to maintain the availability of an echocardiographic examination for all newborns with suspected CCHD. PO has been repeatedly proposed as an addition to PE to fulfill postnatal screening demands [11]. Its affordability and steep learning curve brand it an excellent candidate for use in LMIC settings. 

PO has previously been studied as a screening tool for CCHD. A Dutch study found a sensitivity of CCHD PO screening of 70% by screening 23,959 newborns [12]. Other published articles have evaluated the pooled diagnostic value of CCHD PO screening and reported similar sensitivities [13,14,15]. Apart from Bello et al., these publications have unfortunately not taken PE into their analyses. In a clinical setting, the complete replacement of standard PE by PO is not reasonable, because relative desaturation is less sensitive to duct-dependent systemic defects and non-duct-dependent defects; absolute desaturations (<95%) make the PE screening method also capable of identifying duct-dependent systemic defects and non-duct-dependent defects [16]. The studies that did not take PE into account therefore reported incomplete outcomes and left unexplored screening potential.

The objective of this systematic review and meta-analysis is to determine the optimal practice of physical examination and pulse oximetry as newborn screening methods for critical congenital cardiac disorders. 

## 2. Materials and Methods

This review followed the criteria for reporting systematic literature reviews and meta-analysis as defined by the PRISMA 2020 statement [17]. A completed PRISMA checklist is provided in the Appendix A.

Search strategy: The Embase, PubMed, and Global Health databases were utilized to search for all relevant articles up to 30 November 2023. The search query was specifically designed for Embase and PubMed by using the Emtree and MeSH terms, respectively. The keywords were based on the following subjects: critical congenital heart disease, physical examination, clinical scores, pulse oximetry, and echography. Appendix B contains the complete search queries for all three databases, no filters or limits applied. Additionally, previously published systematic reviews on newborn CCHD screening were scanned for eligible articles outside the bounds of the search query. A PRISMA diagram reflects the selection process [17].

Inclusion and exclusion criteria: The database search results were exported to Rayyan for article selection [18]. No exclusions on the basis of language or study region were made. All titles and abstracts were independently manually screened by two authors (J.T. and N.M.) on the basis of the following inclusion criteria. (1) Screening had to take place in the newborn period and (2) all newborns were screened with both the PE and PO screening method. A CCHD is defined as one of the following pathologies: coarctation of the aorta, double-outlet right ventricle, d-transposition of the great arteries, Ebstein’s anomaly, hypoplastic left heart syndrome, interrupted aortic arch, pulmonary valve atresia, single ventricle, total anomalous pulmonary venous return, tetralogy of Fallot, tricuspid atresia, persistent truncus arteriosus [1]. When information about lesion characteristics was insufficient, they were not classified as critical. For duplicate publications, we selected the most recent and complete versions of reports. Correspondingly, study types such as reviews, systematic reviews, meta-analyses, case reports, case series, conference abstracts, comments, clinical guidelines, and animal studies were excluded. Studies including newborns admitted to the Neonatal Intensive Care Unit (NICU) were also excluded. Disagreements were resolved by consensus and after discussion with a third reviewer (M.G.S).

Data extraction: Two reviewers (J.T. and N.M.) independently extracted information about study characteristics, quality, and test results from each selected article. Every included article was subject to extraction of the following parameters: first author, year of publication, type of study design, sample size, timepoint, location and cut-off values of pulse oximetry measurements, pulse oximetry device, type(s) of physical examination(s), reference standard of CCHD diagnosis, and screening location. For every article selected, the CCHDs previously mentioned were considered to be outcomes of interest. The echocardiogram was used as a reference pattern in all reviews. Notably, in most cases, screens with a positive PE or PO were referred for echocardiography. Most of the reviews did not find any factors that might hinder applicability of the screening, both in terms of selecting the population for the study and screening the reference pattern. 

The combination of the two individual screening methods was evaluated in two ways. A screening is positive if either the PE or PO test is positive or when both were positive. To compare the two combined methods with their individual components, both the PE and PO screening methods were also individually assessed. Table 1 below describes all conceivable outcomes for all four screening methods: PE only, PO only, PE AND PO, and PE OR PO.

For the meta-analysis, the number of True Positives (TP), False Positives (FP), False Negatives (FN), and True Negatives (TN) of all four screening methods were extracted.

Statistical analysis: The diagnostic accuracy was quantitatively defined with specificity, sensitivity, positive Likelihood Ratio (LR+), negative Likelihood Ratio (LR-), and Diagnostic Odds Ratio (DOR). All statistical quantities are accompanied by their respective 95% Confidence Interval (95% CI). The Meta-DiSc software was employed to pool the specificity and sensitivity of all screening methods in forest plots using a random effects model [19]. An assessment of statistical heterogeneity was conducted using the I^2^ statistic calculation describing the total variation percentage among the studies, which may be attributed to heterogeneity and not chance, and an I^2^ more than 75% was considered high. Additionally, Summary Receiver Operating Curves (SROCs) were generated to approximate the Area Under the Curve (AUC). Estimates of sensitivity and false-positive rates were computed and plotted in forest plots according to the predefined subgroups. 

Risk of bias analysis: All included studies were subjected to a Risk of Bias (RoB) and Applicability Concern (AC) assessment with the QUADAS-2 score conducted independently by the authors (J.T. and N.M.) [20]. The four key domains covering patient selection, index test, reference stand (comparator), and flow and timing were evaluated. Any study deemed to have a high risk of bias, scoring ‘high’ for three or more of the four risk-of-bias domains, or concerns regarding applicability, scoring ‘high’ for two or more of the three applicability domains, was excluded from further meta-analyses. For sensitivity analysis, we checked the effect of exclusion of studies according to the Risk of Bias. Subgroup and Meta regression analyses were not applicable.

## 3. Results

Article selection: A total of 2711 articles were screened for inclusion. The majority, 2707 of these articles, were found in the Embase (2070), PubMed (579), and Global Health (58) databases. Four articles were identified through other reviews and eligible articles. After duplicate and title/abstract exclusion, 262 articles remained for detailed assessment. Studies that included only newborns in the NICU or those that did not take PE screening into account were excluded [21,22]; additionally, one study was excluded for inclusion of symptomatic newborns [23]. Subsequently, 20 articles remained for data extraction and meta-analysis. An overview of the search results and selection process is shown in a PRISMA diagram in Figure 1 [17].

Data extraction: Not all included articles provided directly relevant outcomes. Nine articles provided only quantitative CHD data instead of CCHD data. Additionally, designations of CCHDs varied. Some studies defined large ventricular septum defects as critical or defined Ebstein’s anomaly and coarctations as non-critical. For these articles, CCHD data were calculated based on given quantitative or contextual information. Unfortunately, not all articles reported the required data to determine the diagnostic accuracy of all four screening methods and were excluded from further analysis [24,25]. The features of the studies considered were summarized and arranged in a table. Table A1 in Appendix B contains all relevant characteristics of the included studies.

Statistical analysis: Cumulatively, 872,549 newborns with 857 confirmed CCHDs have been screened. This study’s CCHD incidence is 0.98 per 1000 newborns (0.098%). Pooled sensitivity and specificity of all four screening methods are reflected in forest plots and SROCs in Figure 2, Figure 3, Figure 4 and Figure 5.

Figure 2 reflects the diagnostic accuracy of the PE-only screening method. Pooled sensitivity was found to be 0.69 (0.66–0.73 (95% CI)) and specificity was found to be 0.98 (0.98–0.98). The heterogeneity of the sensitivity was considerably high, while the specificity had a low heterogeneity. Additionally, pooled LR+ was 33.9 (22.6–50.7), pooled LR- was 0.4 (0.2–0.7), and DOR was 77.0 (38.5–154.0). The PE-only AUC was determined to be 0.97. 

Figure 3 reflects the PO-only screening method with a pooled sensitivity and specificity of 0.78 (0.75–0.82) and 0.99 (0.99–0.99), respectively. Heterogeneity was assessed moderate for sensitivity and low for specificity. Pooled LR+ was 238.8 (94.5–603.7), pooled LR- was 0.2 (0.1–0.4), and DOR was 1277.4 (391.2–4171.4). The AUC was 0.93.

Figure 4 represents the statistical analysis of the PE AND PO combined method. Sensitivity and specificity were 0.50 (0.45–0.55) and 0.99 (0.99–0.99), respectively. The heterogeneity of sensitivity was determined to be high and for specificity, it was low. Pooled LR+ was 328.9 (49.7–2176.8), pooled LR- was 0.6 (0.3–1.1), and DOR was 691.7 (102.2–4682.5). The AUC was determined to be 0.95.

Finally, Figure 5 characterizes the PE OR PO method. The achieved sensitivity and specificity were 0.93 (0.91–0.95) and 0.98 (0.98–0.98), respectively. Pooled LR+ was 63.6 (37.1–109.1), pooled LR- was 0.1 (0.0–0.3), and DOR was 707.1 (192.1–2602.5). The AUC was approximated to be 0.98. The average Negative Predictive Value (NPV) and Positive Predictive Value (PPV) were 99.99% and 3.53%, respectively. 

Risk of Bias: All studies were subjected to the RoB and AC domains of the QUADAS-2 assessment. The overall score given to each study is equal to the score of the domain with the highest risk. Once the application of the quality criteria was performed, twenty articles were selected for data extraction; one article was excluded after conducting the quality assessment, as it did not include comparative data between the physical examination and oximetry [26]. Figure A1 in Appendix C displays the individual scores for every domain and study. A summary of all scores is shown in Figure A2 in Appendix C.

Sensitivity Analysis: The sensitivity Analysis characterizes the low RoB in the PE OR PO method. Pooled sensitivity was found to be 0.94 (0.90–0.96 (95% CI)) and specificity was found to be 0.96 (0.96–0.96). The AUC was determined to be 0.98. The pooled overall sensitivity of PE or PO screening method did not change significantly with the exclusion of studies exhibiting high Risk of Bias (Figure 6).

Heterogeneity Test of the Meta-Analysis and Sensitivity Analysis: The sensitivity I^2^ = 82.70 showed significant heterogeneity; *p* value = 0.0. Sensitivity analysis was conducted according to Risk of Bias. It showed that the heterogeneity had a significantly lower I^2^ of 43% (*p* value of 0.11). 

Publication Bias: A funnel plot was used to determine publication bias. The funnel plot was asymmetric, and Begg’s test was significant (*p* = 0.001), indicating the presence of publication bias (Appendix A).

## 4. Discussion

Sensitive screening of newborns for CCHDs is imperative for reducing undiagnosed discharges. Since PO has been proposed to improve CCHD screening, its general diagnostic value has been sought after. However, its additive value to the existing screening mode, PE, has yet to be evaluated recently. This meta-analysis studied the cumulative diagnostic value of both individual modes of screening and their synergetic diagnostic potential.

The PO-only screening method yields the highest DOR at 1277 with a sensitivity and specificity of 0.78 and 0.99, respectively. The pooled sensitivity is moderate, causing the SROC curve to provide a lower AUC (0.93) than that of the PE OR PO combined screening method (AUC of 0.98). Although the specificity is lower than PO only, the PE OR PO method provides a better combination of sensitivity and specificity (0.93 and 0.98, respectively) because its sensitivity of 0.93 is the highest of all four methods. An effective screening method is to provide sufficient specificity to cope with the strain on the availability of echocardiography. However, the potential of the screening method to identify diseased patients, thus the sensitivity, is deemed more important.

Bello et al. published a comparable systematic review and meta-analysis in 2019 [15]. With five included studies, they found a comparable pooled sensitivity and specificity at 0.92 and 0.98, respectively. By including newly published articles, this meta-analysis quadruples the number of included studies and more than doubles the number of cumulatively screened newborns.

A larger meta-analysis was published in 2017 by Du et al. about the role of PO-only screening for CCHDs [13]. Their inclusion of 22 articles resulted in a pooled sensitivity of 0.69 and a pooled specificity of 0.99. Although just a small portion of studies were shared between Du et al. and this study, the results correspond well with our yielded PO-only sensitivity and specificity of 0.78 and 0.99, respectively.

Ma et al. have described successfully implementing the combined screening method in Shanghai [45]. For CCHDs, they recorded a perfect sensitivity of 1 and a specificity of 0.98, resulting in a general downtrend of the overall infant mortality rate. Their pilot studies, which reported data on single and combined methods, have been included in this analysis [27,28].

The additive diagnostic value of PO for standard PE-only screening is demonstrated by the combined sensitivity. The individual sensitivities of 0.78 and 0.69, respectively, add up to a sensitivity of 0.93 of the PE OR PO method. This implies that some of the CCHD cases not identified by PE were identified by PO. This mechanism also provides an explanation of why the PE AND PO screening method yields a far lower sensitivity of 0.50 compared to its PE OR PO counterpart. Our findings emphasize the importance of this screening strategy in that utilizing both PO and PE in clinical practice is an excellent strategy to help detect cases that might have been missed by PE alone, leading to a reduction in the rate of false negative results. Since the PO-only screening strategy is less sensitive to duct-dependent systemic defects and non-duct-dependent defects, the PE screening method is capable of identifying duct-dependent systemic defects and non-duct-dependent defects [16]. This is a crucial aspect for accurate diagnoses. The findings are consistent with what has been previously reported in the literature, indicating that the combination strategy has improved sensitivity and allows for the timely detection of 30 additional cases per 1000 live births that would be missed if PE was used alone [8,22].

To improve the detection of all types of CCHDs, 12 of the 20 included studies measured pre-ductal oxygen saturation on the right arm and post-ductal oxygen saturation on either leg (see the ‘Location of PO sensor’ column in Table A1). Differences above 3% between the arm and leg measurements could indicate duct-dependent pulmonary defects. The American Academy of Pediatrics has suggested using both extremities including the use of the difference of more than 2% instead of 3% between extremities in their new guideline for screening newborns CCHD [46]. This recommendation is based on the limitations of previous studies that predominantly used post-ductal measurements, which found no significant difference in accuracy when one or two limbs were used in the screening process [14,29]. While there have been conflicting results regarding the best method for accuracy and simplicity of the one-limb screening method, the use of both extremities and the difference of 2% ensure that the vast majority of CCHD cases are detected [46,47]. A cut-off point of 95% is generally accepted, and all included studies in our review used this cut-off point. 

Cumulatively, 872,549 newborns with 857 CCHDs have been screened (see the ‘Total’ row in Table A1). This accounts for an incidence close to 1 per 1000 newborns (0.098%). All included studies consistently report about half of the incidence of CCHD that is reported in literature [9,48]. The cause of this effect lies within the individual study exclusion criteria and the exclusion criteria of this analysis. Most studies excluded prenatal diagnoses, premature newborns, and newborns with dysmorphic features (see the ‘Study group’ column in Table A1). This effect is strengthened by our decision to exclude NICU-only studies. Our outcomes of sensitivity and specificity are independent of incidence, but the confidence interval of sensitivity is not. A more representable incidence would have allowed for narrower confidence intervals, and thus a better approximation of the true sensitivity. The heterogeneity of the sensitivities is also found to be much more substantial than the heterogeneity of the specificities for all four screening methods. This is a plausible result of the large confidence intervals. This discrepancy in heterogeneities is also found in the comparable meta-analyses of Bello et al. and Du et al. [13,15]. For the PE OR PO screening method, most sensitivity confidence intervals do overlap, which reduces the heterogeneity to moderate.

Chen et al. and Vaidyanathan et al. did not report the required data to manually convert their results to CCHDs [30,31]. Instead, major CHDs were included in the meta-analysis. Major CHDs are CCHDs plus defects such as large ventricular and atrial septum defects. The diagnostic value of the Chen et al. screening appears to be in accordance with the other included studies. However, the sensitivity of the PO-only screening method of Vaidyanathan et al. is much lower than its peers. Its authors account for this on the low number of cyanotic inclusions and poor performance/physical placement of the pulse oximeters [31]. As Vaidyanathan et al.’s included newborns only account for 0.6% of all screenings, its weight on the pooled results is considered minimal. However, Vaidyanathan et al. underline that proper protocolization of the screening method and training for medical personnel is imperative for its accuracy.

Clear guidelines and regular training are essential for performing the PE correctly due to its absence of quantification and thus inherent subjectivity [49]. Symptoms such as cyanosis and tachypnoea were frequently studied as part of the PE screening. Some studies also mentioned assessing the quality of the femoral pulsations (see the ‘PE components’ column in Table A1). All included papers elected cardiac auscultation as part of the PE screening. Notably, the absence of murmurs in newborns does not exclude a CCHD. Such murmurs can be difficult to detect or may not even be present in the newborn period as in the case of transposition of great arteries. Discrimination between common innocent physiological murmurs and pathological murmurs is vital for diagnostic screening accuracy. Diagnostic specificity of all murmurs has been found to be just 0.38 for newborns [50,51]. Not all included studies explicitly state whether they included all murmurs or just pathological murmurs, such as one from an aortic coarctation determined to be a positive screen. However, a clue about the employed methods is given in Figure 2B, where the PE specificity is generally high with the exception of Minocha et al. [32]. This study explicitly stated not to discriminate between murmurs and designated all newborns with a murmur as positively screened. All other studies, with a similar high PE specificity, likely did discriminate between physiological and pathological murmurs. Minocha et al.’s study was therefore allotted a substantial risk of bias in domain 1 (D1) of the QUADAS-2 score: patient selection (Figure A1 and Figure A2). Various clinical scoring systems have been developed to enhance the clinical detection of CHD, each having different sensitivity and specificity levels [52]. The NADAS Criteria have been demonstrated to have high accuracy and are therefore considered superior among these scoring systems [53,54]. As a result, the NADAS Criteria can be employed when screening newborns for heart diseases.

Lastly, it must be noted that positively screened newborns with the PE OR PO screening method have an average chance of 3.5% (PPV) of having a CCHD. This does not imply that the remainder of positively screened newborns (96.5%) are healthy. Although this study did not report any other diseases, newborns with failed screenings have a higher chance of a non-critical CHD, an infection, or a respiratory pathology. Narayen et al. assessed newborns with a PO-only screening and found that 61% of the false positives had noncardiac illnesses [12].

Recommendations: One study in particular, Uygur et al., did not only assess PE and PO as a screening method for CCHD but also employed the Peripheral Perfusion Index (PPI) [33]. The PPI is a measure of tissue perfusion, which is independent of oxygen saturation. Most pulse oximeters already incorporate a PPI mode to assess the signal quality of the saturation measurement. Uygur et al. exposed the shortcomings of the PO-only screening by reporting two false negative cases of coarctations of the aorta. A coarctation can be missed by the PO-only screening due to its acyanotic nature. The two cases were identified by PPI because lower extremity tissue perfusion was insufficient due to the constricting aorta. Recently, Jiang et al. performed a meta-analysis on this topic, reaffirming the encouraging findings of Uygur et al. [55]. The potential of PPI to the fill the remaining sensitivity gap with little practical protocol modification is very promising. An extension of the screening age to 6 weeks could potentially also increase sensitivity for less symptomatic newborns at birth.

Limitations: The main limitation of the included studies is reflected in domain 4 (D4) of the QUADAS-2 score: flow and timing (Figure A1 and Figure A2). Subjecting a large group of negatively screened newborns to postnatal echocardiography is not always feasible for studies. Except for Saxena et al., the included studies did not refer all screened newborns for an echocardiography [34]. Only screens with either a positive PE or PO were referred. This form of bias causes a decline in false negatives and subsequently falsely projects a higher sensitivity. To counteract this effect, nine studies introduced a follow-up after discharge. By telephoning the parents and asking about any symptoms after discharge, false negatives can be identified. In other cases, falsely negative-screened newborns became symptomatic after discharge and presented to the emergency room or the morgue after discharge. 

This review has several strengths, including a comprehensive search of recent literature and a standardized quality assessment of the articles included. However, one of the limitations is that there were few high-quality studies among those included. The asymmetry in the funnel plot List of Abbreviations suggested the presence of publication bias. Although the low risk of bias analysis in Figure 6 did not differ notably, the presence of publication bias has to be taken into account when interpreting results. Nevertheless, no studies with a high risk of bias were added.

## 5. Conclusions

The objective of this systematic review and meta-analysis is to determine the optimal practice of PE and PO as newborn screening methods for CCHD. Pooled sensitivity and specificity determined that combining both screening methods improved both statistical quantities to 0.93 and 0.98, respectively. The considerable magnitude of screened newborns and the topographical diversity of studies encourage study results to be applied in both HIC and LMIC settings. LMICs which do not have a standard fetal ultrasound screening protocol could potentially improve CCHD detection the most, including the ability to detect major shunt lesions early when incorporating PE in a systematic manner. Adoption of the combined screening method is recommended.

## List of Abbreviations

ACApplicability Concern AUCArea Under the Curve CHDCongenital Heart Disease CCHDCritical Congenital Heart DiseaseCIConfidence IntervalDORDiagnostic Odds Ratio FN False Negatives FP False PositivesHICHigh-Income CountriesJ.TJari TristanJ.T.v.V.Jari Tristan van VlietLMICLow Middle Income CountriesLR-Negative Likelihood Ratio LR+Positive Likelihood Ratio M.G.SMartijn G. SliekerN.G.M.Naizihijwa Gadi MajaniN.MNaizihijwa MajaniNICUNeonatal Intensive Care UnitNPV The average Negative Predictive Value P.CPilly ChilloPEPhysical ExaminationPO Pulse OximetryPPV Positive Predictive Value PRISMAPreferred Reporting Items for Systematic Reviews and Meta-AnalysesQUADASQuality Assessment of primary Diagnostic Accuracy StudiesRoB Risk of BiasSROCSummary Receiver Operating Curves TN True NegativesTP True Positives

## Figures and Tables

**Figure 1 children-11-00047-f001:**
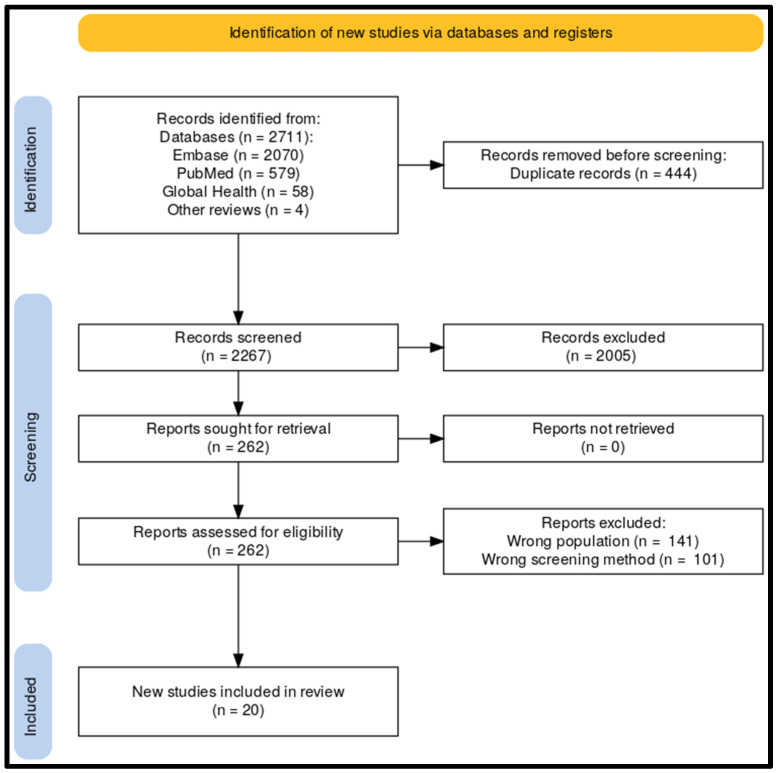
PRISMA diagram of article selection.

**Figure 2 children-11-00047-f002:**
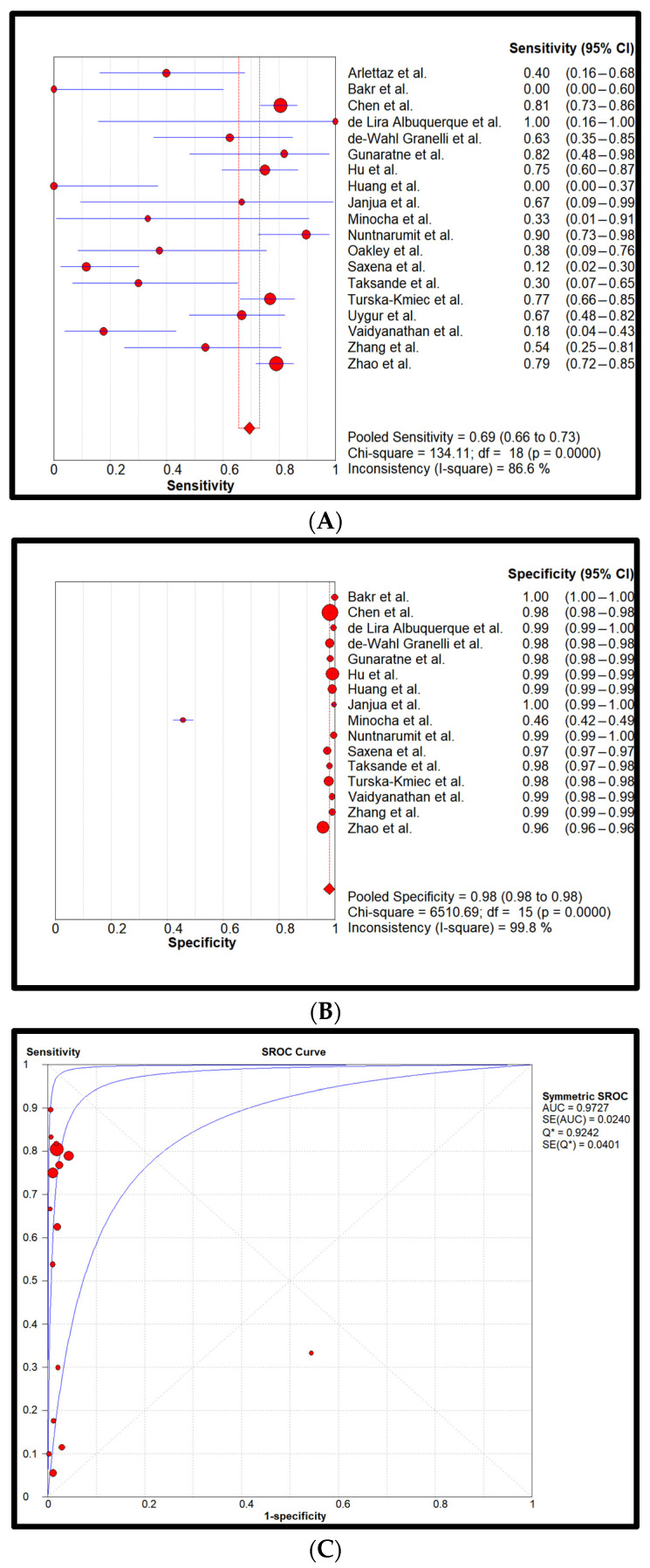
Pooled sensitivity (**A**), pooled specificity (**B**), and the SROC curve (**C**) of the PE-only screening. The 95% CI of the middle blue SROC curve is represented by the two outer blue curves in subplot C [26,27,28,29,30,31,32,33,34,35,36,37,38,39,40,41,42,43,44].

**Figure 3 children-11-00047-f003:**
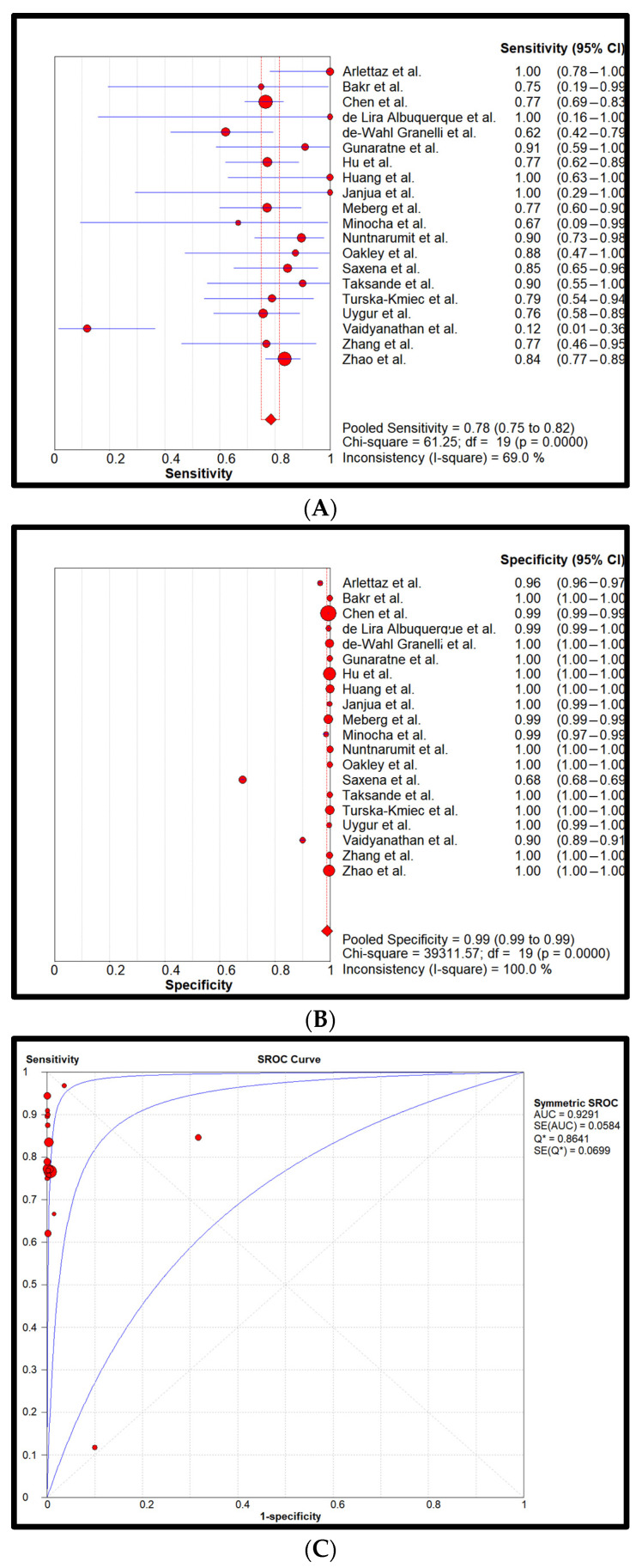
Pooled sensitivity (**A**), pooled specificity (**B**), and the SROC curve (**C**) of the PO-only screening. The 95% CI of the middle blue SROC curve is represented by the two outer blue curves in subplot C [26,27,28,29,30,31,32,33,34,35,36,37,38,39,40,41,42,43,44].

**Figure 4 children-11-00047-f004:**
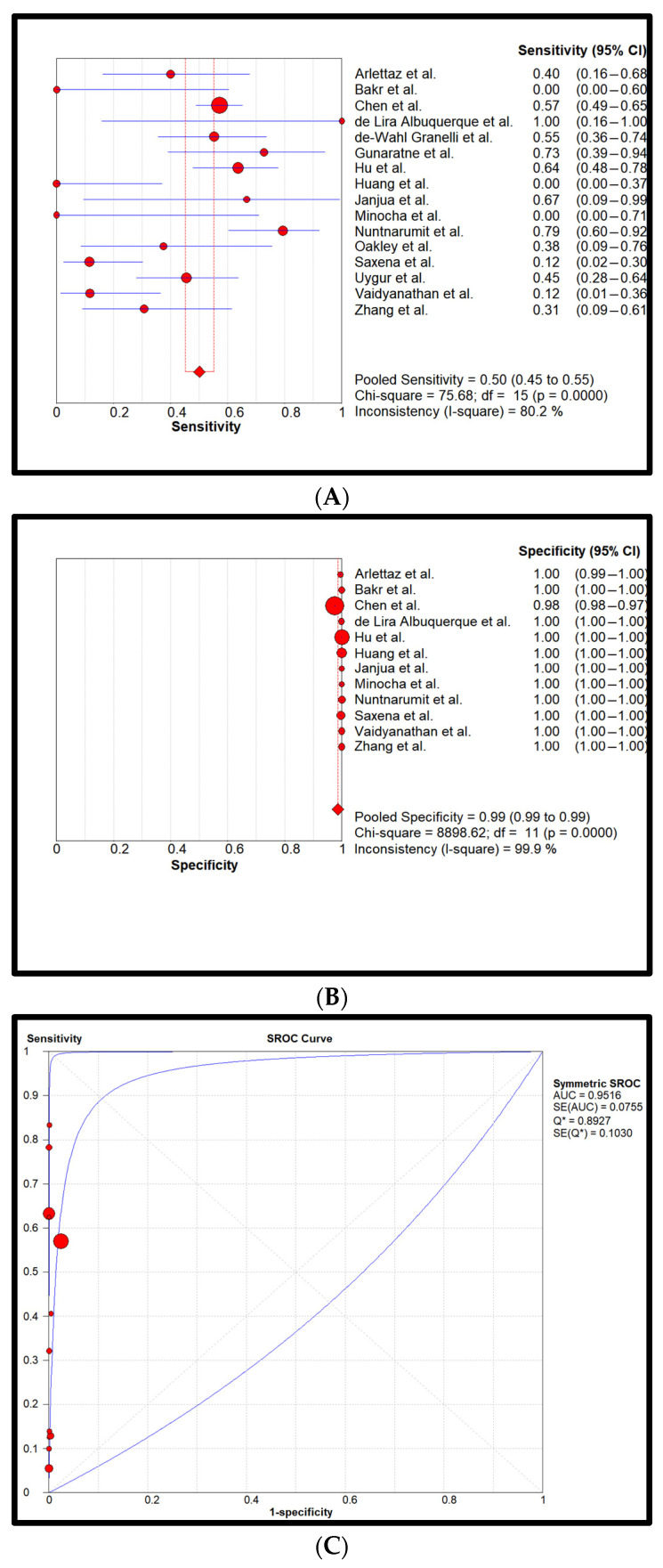
Pooled sensitivity (**A**), pooled specificity (**B**), and the SROC curve (**C**) of the PE AND PO screening. The 95% CI of the middle blue SROC curve is represented by the two outer blue curves in subplot C [26,27,28,29,30,31,32,33,34,35,36,37,38,39,40,41,42,43,44].

**Figure 5 children-11-00047-f005:**
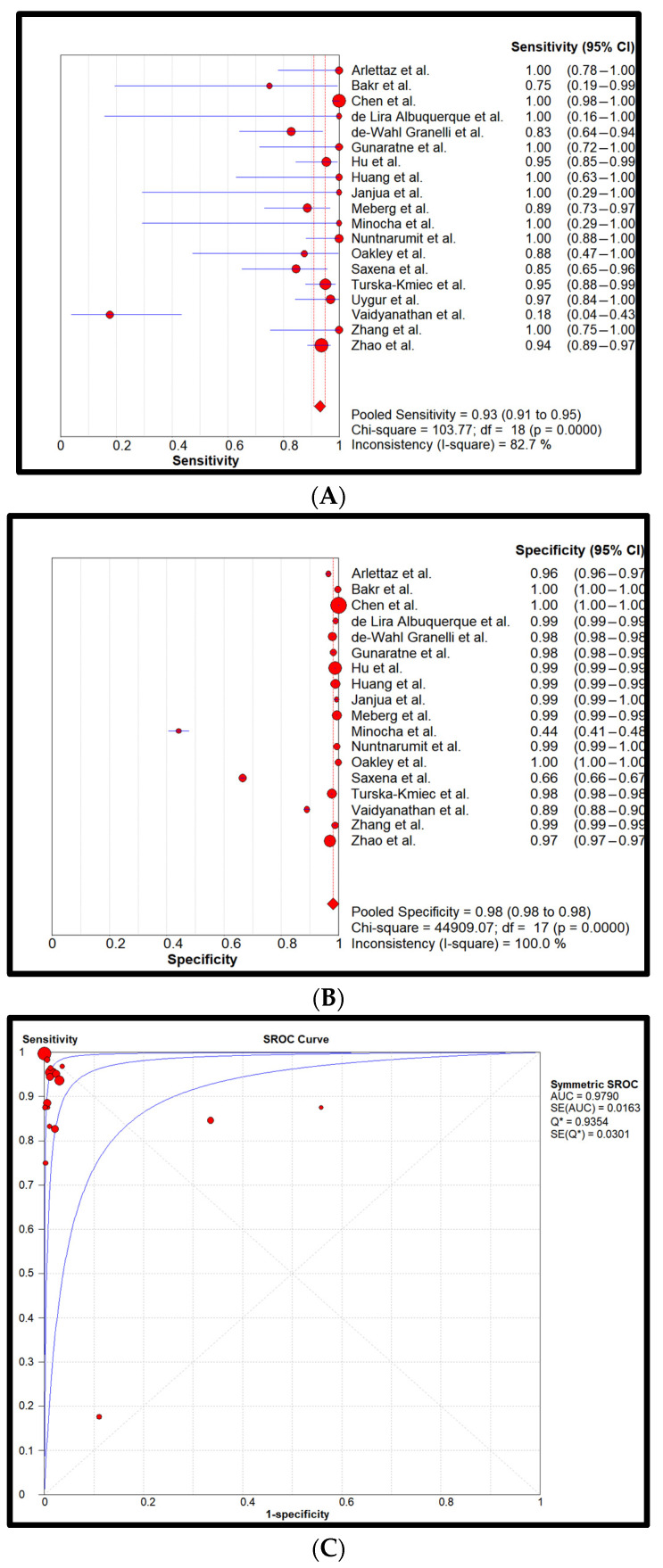
Pooled sensitivity (**A**), pooled specificity (**B**), and the SROC curve (**C**) of the PE OR PO screening. The 95% CI of the middle blue SROC curve is represented by the two outer blue curves in subplot C [26,27,28,29,30,31,32,33,34,35,36,37,38,39,40,41,42,43,44].

**Figure 6 children-11-00047-f006:**
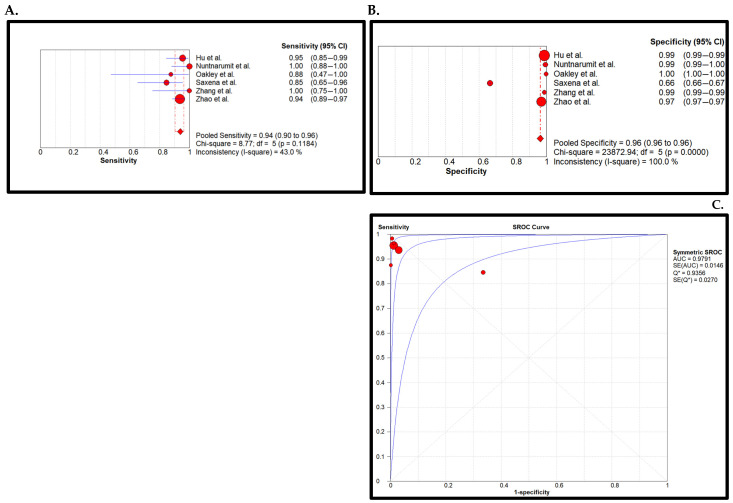
Low Risk of Bias pooled sensitivity (**A**), pooled specificity (**B**), and the SROC curve (**C**) of the PE OR PO screening. The 95% CI of the middle blue SROC curve is represented by the two outer blue curves in subplot C [27,28,34,40,41,44].

**Table 1 children-11-00047-t001:** All hypothetical PE and PO test outcomes for a newborn are in the two left columns. Their respective screening outcomes for all four methods are in the four right columns. Zero is negative and one is positive.

PE	PO	PE Only	PO Only	PE AND PO	PE OR PO
0	0	0	0	0	0
0	1	0	1	0	1
1	0	1	0	0	1
1	1	1	1	1	1

## Data Availability

Data are contained within the article and Appendix A.

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
