# Peer review of "Diagnostic Accuracy of Physical Examination and Pulse Oximetry for Critical Congenital Cardiac Disease Screening in Newborns"

_children, 2023, doi:10.3390/children11010047_

Round 1

Reviewer 1 Report

Comments and Suggestions for Authors

The review is comprehensive, but the general aspect could be improved with some changes hereby I suggest:

Material and methods

Page 2, line 85-89

Please specify if atresia of the pulmonary valve was considered as critical congenital heart disease.

Results

Please specify what type of congenital heart diseases were diagnosed (for each study) and if is possible (if available data) what type of congenital heart disease is more likely to be diagnosed using the combined screening.

References

Please correct the following references according to the instructions (17, 18, 19, 24, 26, 33, 35, 36, 37, 39, 40, 42).

Author Response

Dear Reviewer,
Thank you for giving us the opportunity to submit a revised draft of our manuscript titled ‘Diagnostic Accuracy of Physical Examination and Pulse Oximetry 
for Critical Congenital Cardiac Disease Screening in Newborns’ to Children’s Journal. We appreciate the time and effort you have dedicated to providing 
valuable feedback on our manuscript. We are very grateful to you for your insightful comments on our paper. We have been able to incorporate changes to 
reflect most of the suggestions provided by the reviewers. Here is a point-by-point response to your comments and concerns.
Sincerely,
Martijn, Slieker
Correspondent Author

Reviewer 2 Report

Comments and Suggestions for Authors

This sistematic review focuses on the sensitivity and specificity of 2 easy to apply manouvers for the diagnosis of critical congenital heart disease; clinical examination and pulsoximetry.

The articles have screened an impressive number of articles for inclusion in their metananalysis.

Discussions: Please provide an explanation to the fact that “PE OR PO method provides a better combination of sensitivity and specificity”. I understand the values you have obtained for specificity and sensitivity, but I do not see the logical explanation of this conclusion. This also contradicts the actual conclusion of the study.

In the murmurs discussion, please also take into consideration that some critical congenital heart defects, such as transposition of great arteries with intact ventricular septum may not be accompanied by a murmur.

The conclusion of the study is already largely known to clinicians. Please elaborate in terms of the new contribution your article is bringing.

Author Response

(The authors gave the same response as above.)

Reviewer 3 Report

Comments and Suggestions for Authors

The authors have rightly focussed on the examination findings in the newborn as a pointer to the presence of congenital heart disease (CHD). Providing references to support that observation would be helpful. To the physical examination has been added the additional gain by carrying out neonatal pulse oximetry, with data to support that finding. It would be helpful for the readership for you to clearly state what are the minimal requirements for a cardiovascular examination in the newborn and if not carried out what you recommend to be the minimal requirements. To that should be added the technique and the parameters of neonatal pulse oximetry rather than relegate that to your Discussion (line 209 – 216). Providing such information early on will orientate the readership to what is being discussed together with their importance.

You have commented in your Introduction that prenatal diagnosis is limited to “high income countries”. You go on to describe home deliveries etc. The question then arises as to whether you would expect nurses, midwives, or lay personal to be able to examine the newborn meaningfully. At times general practitioners and/or obstetricians/paediatricians have difficulty with their examinations being incomplete or inadequate with difficulty in interpreting their findings. No comment has been made of that issue.

Your Introduction includes a list of critical congenital heart disease but fails to distinguish between those that require urgent intervention while others which may be coasted along but may eventually require intervention. The former should include all newborns with suspected cyanotic congenital heart disease. The latter will include such conditions as a total anomalous pulmonary venous drainage unless obstructed. No comment has been made about large communication between the circulations, for example a large VSD or patent ductus arteriosus which may also give rise to significant morbidity and if untreated mortality if unrecognised. Those lesions generally are not picked up during the newborn period and will not be recognised by the pulse oximetry analysis. You may wish to make that point within your Introduction or at least under your Limitations as Dr Majani well understands. Comments about a duct dependent pulmonary and/or systemic circulations would also be helpful so as to allow the readership to understand what it is that the pulse oximetry hopes to recognise. You have partially eluded to the latter.

The Figures included in the Results are too detailed for the body of the paper and may well be shifted to an Appendix. The type needs to be enlarged to allow for easier reading. Is there any way for you to present that information in a pictorial manner?

Your Limitations are far too long and include a further review of the literature which is inappropriate under that heading. Limitations should concentrate on the problems associated with your study that you are describing with possible suggestions as to how to address them in future studies maybe at the end of your Conclusions.

Minor points:

-          A small box incorporating the many abbreviations you use will be helpful.

-          Echography (line 20 and elsewhere) should be changed to echocardiography

Author Response

Dear Reviewer,

Thank you for giving us the opportunity to submit a revised draft of our manuscript titled ‘Diagnostic Accuracy of Physical Examination and Pulse Oximetry for Critical Congenital Cardiac Disease Screening in Newborns’ to Children’s Journal. We appreciate the time and effort you have dedicated to providing valuable feedback on our manuscript. We are very grateful to you for your insightful comments on our paper. We have been able to incorporate changes to reflect most of the suggestions provided. Here is a point-by-point response to your comments and concerns.

Sincerely, Martijn, Slieker

Correspondent Author

Round 2

Reviewer 2 Report

Comments and Suggestions for Authors

Thank you for adressing my concerns.